# Effect of Biodegradable Coatings on the Growth of *Aspergillus flavus* In Vitro, on Maize Grains, and on the Quality of Tortillas during Storage

**DOI:** 10.3390/molecules27144545

**Published:** 2022-07-16

**Authors:** Rosa I. Ventura-Aguilar, César Gónzalez-Andrade, Mónica Hernández-López, Zormy N. Correa-Pacheco, Pervin K. Teksür, Margarita de L. Ramos-García, Silvia Bautista-Baños

**Affiliations:** 1CONACYT-Centro de Desarrollo de Productos Bióticos, Instituto Politécnico Nacional, Carretera Yautepec-Jojutla, km. 6.8, San Isidro, CEPROBI 8, Yautepec Morelos 62731, Mexico; riventuraag@conacyt.mx; 2Facultad de Nutrición, Universidad Autónoma del Estado de Morelos, Calle Iztaccihuatl S/N, Col. Los Volcanes, Cuernavaca Morelos 62350, Mexico; jingsible@gmail.com; 3Centro de Desarrollo de Productos Bióticos, Instituto Politécnico Nacional, Carretera Yautepec-Jojutla, km. 6.8, San Isidro, CEPROBI 8, Yautepec Morelos 62731, Mexico; monibrisa@hotmail.com (M.H.-L.); zormynacary@yahoo.com (Z.N.C.-P.); 4Department of Plant Protection, Ege University, Erzene Mahallesi Ege Üniversitesi Merkez Yerleşkesi, Bornova, 35040 İzmir, Turkey; pervin.kinay@ege.edu.tr

**Keywords:** *Zea mays* L, chitosan, propolis, pine resin extract, disease incidence: aflatoxins, nanocoatings

## Abstract

The fungus *Aspergillus flavus* causes serious damage to maize grains and its by-products, such as tortilla. Currently, animal and plant derivatives, such as chitosan and propolis, and plant extract residues, respectively, are employed as alternatives of synthetic fungicides. The objective of this research was to evaluate the efficacy of several formulations based on propolis-chitosan-pine resin extract on the in vitro growth of *A. flavus*, the growth of maize grain plantlets and the quality of stored tortillas at 4 and 28 °C. The most outstanding formulation was that based on 59.7% chitosan + 20% propolis nanoparticles + 20% pine resin extract nanoparticles; since the in vitro conidia germination of *A. flavus* did not occur, disease incidence on grains was 25–30% and in tortillas, 0% infection was recorded, along with low aflatoxin production (1.0 ppb). The grain germination and seedling growth were markedly reduced by the nanocoating application. The percentage weight loss and color of tortillas were more affected by this coating compared to the control, and the rollability fell within the scale of non-ruptured at 4 °C and partially ruptured at 28 °C. The next step is to evaluate the toxicity of this formulation.

## 1. Introduction

There are phytopathogenic fungi that, in addition to altering the quality of the horticultural commodity, produce toxic compounds, known as mycotoxins. In the case of the *Aspergillus* species, *A. flavus* principally produces the mycotoxin aflatoxin which causes serious health issues for the humans and animals that consume it [1]. This fungus exists on an extremely large range of agricultural hosts including various fruit, tree nuts, cereals, and mainly stored grains such as maize [2,3].

The conidia of *A. flavus* enter through the pedicel or from a wound in the maize grain and germinate inside, where the hyphae begin to grow after 3 to 7 days. Later, the fungus generates conidiophores that release conidia to contaminate other maize grains. The optimal temperature range for its development is between 10 and 55 °C. Aflatoxins develop once the fungus begins to develop conidiophores, but they also occur when the fungus feels stressed, either by a competitor, lack of nutrients and water, or the presence of chemical fungicides [1].

Maize is a product consumed heavily in Mexico, due to its high nutritional content and versatile use in a wide variety of foods. One of the by-products derived from maize with the highest level of consumption in the country is the tortilla. However, as reported by Martínez et al. [3], in Mexico, tortilla can be contaminated by *A. flavus*. The authors stated that in various maize producing states of the country, this fungus is responsible for crop losses of up to 50%. Additionally, grains infected with *A. flavus* can also be contaminated with high doses of aflatoxins [4]. For example, Mendez-Albores et al. [5] reported aflatoxin content in the range of 2 to 9 ppb and 6 to 36 ppb in tortillas, made using the traditional and ecological making processes, respectively. Furthermore, Anguiano-Ruvalcaba et al. [6] reported that in the state of Tamaulipas, the maize from the field exhibited concentrations of 45–65 mg of AFB L/kg, and after being stored for two months in conditions of high temperature and humidity, the concentrations exceeded 250 mg of aflatoxin B1/kg. Added to this, the lack of adherence to the NOM-247-SSA1-2008 (Norma Oficial Mexicana 2008) [7] makes the control of aflatoxins difficult to follow.

To solve this problem, various approaches have been tested to control *A. flavus*, the most common method being the use of chemical products such as, among others, imidazoles, thiabendazole, and sodium o-phenylphenate [8]. However, aside from being expensive, these synthetics that can generate strains resistant to fungicides after several years of exposure exert phytotoxic effects on grain germination, and restrict maize imports and exports [3,9]. Alternative methods include agronomic practices [10], biological control with the antagonist *Bacillus* [11], and application of natural compounds including, among others, plant extracts, essential oils [12,13], and animal derivatives (chitosan and propolis) [14,15].

With respect to natural compounds, it has been demonstrated that formulations alone or combined based on propolis (a resinous compound created by bees that is rich in active compounds such as flavonoids, phenolic acids, and terpene derivatives) [16], chitosan (a polysaccharide obtained from partially deacetylated chitin) [17,18], and pine resin extracts (a plant secretion from tree bark, particularly of conifers) [19] provided an effective control over the development of *A. flavus* in vitro and of various horticultural products artificially infected by this fungus, including ficus and tomato [15,20,21]. In all of these studies, not only was the incidence of *A. flavus* disease notably reduced compared to the untreated ones but also the production of the aflatoxins was remarkably low with corresponding values less than 20 ppb.

In agriculture, nanotechnology has shown great potential for the development of new technologies. Applications include, among others, the production and development of food processing systems, chemicals (fertilizers, herbicides), and growth regulators. Currently, nanotechnology has also focused on the application of new and natural compounds that reduce or control the incidence of diseases caused by fungi during the postharvest storage of horticultural commodities [22].

To increase its antimicrobial capacity, the nanotechnology can be used with the advantage that some of the above-mentioned natural compounds can be encapsulated, as it has been proven that nanoparticles provide a larger contact surface, greater dispersion, and better conservation of the active product [23,24]. For instance, significant effects have been demonstrated in controlling the growth of *A. flavus* with nanostructured formulations based on propolis at 1.2% and chitosan, achieving an inhibition of c.a. 40% at 10 days of incubation [15].

For these reasons, the objectives of this research were to evaluate the efficacy of formulations based on propolis-chitosan-pine resin for: (1) the growth of *A. flavus* on nutrient media; (2) the incidence of *A. flavus* on maize grains and its effect on grain growth; and (3) the incidence of *A. flavus* on maize tortillas, storage quality, and the production of aflatoxins.

## 2. Results

### 2.1. Effect of Natural-Based Coatings on In Vitro A. flavus Development

For these studies, there were significant differences (*p* < 0.001) among the treatments of the variables: mycelial growth and spore germination. The mycelial growth of *A. flavus* was affected by the coatings containing chitosan + pine resin extract (T7), chitosan + propolis + pine resin extract (T9), and chitosan + nanoparticles of propolis and nanoparticles of pine resin extract (T10) (Table 1, Figure 1a). Compared to the control, the highest inhibition of *A. flavus* was in nutrient media with chitosan + propolis + pine resin extract (T9) with a corresponding inhibition value of approximately 75%. With respect to spore germination, this was completely deterred throughout the whole incubation period when treated with chitosan + nanoparticles of propolis + nanoparticles of pine resin extract (T10) (Figure 1b). By contrast, the spore germination with the remaining treatments reached approximately 73% at the end of the incubation period of 10 h.

### 2.2. Effect of Natural-Based Coatings on Grain Maize Disease Incidence, Germination, and Plant Growth

Overall, the highest grain maize infection levels were recorded on the non-disinfected rather than the disinfected grains (Figure 2a,b). In both cases, there were significant differences (*p* < 0.001) among treatments. *Aspergillus flavus* infection began from the first day of incubation up until the end of the 7 days incubation. The lowest disease infection corresponded to the untreated and non-inoculated grains (C2) and in the coated ones with chitosan + nanoparticles of propolis + nanoparticles of pine resin extract (T10) with infection values of approximately 25–30% (non-disinfected and disinfected grains, respectively). The highest grain infection of approximately 60% was in the grains treated with chitosan + pine resin extract (T7) and in the inoculated and non-disinfected grains (TC1).

Regarding grain germination, there were significant differences (*p* < 0.05) among treatments at the 3rd and 4th days of incubation (Figure 3a). The germination began after three days in all treatments and in general, the percentage germination was around 22%. With respect to radicle and seedling development, there were significant differences among treatments (*p* < 0.05) (Figure 3b). In this case, both the highest radicle and seedling growth was evident in the grains treated with the coatings containing chitosan + propolis + pine resin extract (T9) (7.0 and 4.5 cm, respectively) and the chitosan + pine resin extract coating (T7) (5.0 and 4.0 cm, respectively).

### 2.3. Effect of Natural-Based Coatings on the Quality of Tortilla Made with Coated Grains

There were significant differences (*p* < 0.05) in all variables relating to tortilla quality during sampling at both storage periods. With respect to humidity, the highest loss was in tortillas of the coating-based chitosan + pine resin extract (T7) treatment with a corresponding value of 25%, and the lowest in tortillas made with grains treated with nanoparticles of propolis + nanoparticles of pine resin extract (T10) with a corresponding value of approximately 20% (Figure 4a,b). Except for the control treatment, the weight loss of tortilla increased as the storage period at 4 °C increased, with the lowest values (0.18%) in the control treatment (TC2), while for tortillas treated only with chitosan + pine resin extract treatment (T7) the highest weight loss (0.50%) was shown from the third day of storage (Figure 5a). When tortillas were stored at ambient temperature, the highest weight loss values of approximately 1% was in tortillas treated with chitosan + propolis + pine resin extract (T9) and the lowest (0.30%) in the untreated tortillas (Figure 5b).

Figure 6a–d depicts the results of the rollability variable. In this case, the treatment with propolis + pine resin extract (T9) stored at 4 and 28 °C for 1 and 21 days resulted in a tortilla rupture within the scale 1–3 (partially ruptured) while, except for the treatment with chitosan + pine resin extract (T7), the remaining ones registered 0 on the scale (unruptured tortilla).

During the 21 days of storage, there were color changes for tortillas at 28 °C of all treatments (Figure 7). The highest color changes were in tortillas made with grains previously treated with chitosan + nanoparticles of propolis + nanoparticles of pine resin extract (T10) and the untreated ones (TC2).

With respect to the incidence of *A. flavus* on tortillas, there was no infection during the entire given storage period when the grains were previously treated with chitosan + nanoparticles of propolis + nanoparticles of pine resin extract (T10). During controlled temperature and at the beginning of acclimation period, no infection was detected with the treatment with chitosan + propolis + pine resin extract (T9); however, after 3 days’ storage infection levels reached 100%. The non-treated tortillas made with inoculated grains (TC1) had the highest infection levels up to the seventh day of storage (Table 2). Regarding aflatoxin production, overall, low levels were detected at 0 and 21 days of storage for all treatments (Table 3) but the lowest levels of aflatoxins were in tortillas made with grains treated with chitosan + nanoparticles of propolis + nanoparticles of pine resin extract (T10) at 0 and 21 days (0.9 and 1.0 ppb, respectively) storage, while the highest levels corresponded to the inoculated grains and non-coated (TC1) in both storage periods (2.4 and 3.2 ppb).

## 3. Discussion

In this study, a fuller picture of the effects of applying other alternatives that included natural products to deter *A. flavus* infection on maize and on the subproduct tortilla was obtained. Additional information was also generated regarding the response of the treated grains on seedling development.

The three natural products tested alone or combined were: chitosan, propolis, and pine resin extract. The results indicated that the coating that stood out the most was that based on 59.7% chitosan + 20% propolis nanoparticles + 20% pine resin extract nanoparticles (T10). This was due to its inhibition of the spore production of *A. flavus* (in vitro studies), to the inhibition of this fungus on grain maize and on tortillas made with the coated grains with this formulation, along with the lowest production of extremely toxic aflatoxins during storage at 4 °C. The results demonstrated that these coatings based on natural products controlled *A. flavus* development with no adverse effects on the overall quality of the tortillas made with the coated grains.

In previous studies carried out in our laboratory on two important agricultural products—figs and tomato—it was demonstrated that when formulations of these natural products were applied, there were similar positive effects with respect to reducing *A*. *flavus* growth and its production of aflatoxins. For example, in fig fruit, the growth inhibition of this fungus was of approximately 20% to 30% under laboratory and semi-commercial conditions, respectively, while for tomatoes, the severity index was about 1.3 cm.

Similar to our findings, in the above studies on figs and tomatoes, the final content of aflatoxins was considerably less than 20 ppb, a value permitted for agricultural edible products by the Food and Drug Administration [25]. In addition, there were no negative effects during fruit ripening, nor on the sensory and nutritional quality of figs and tomatoes [20,21].

To the best of our knowledge, no previous studies have reported the application of coatings on maize grain and its effects on one of its subproducts, i.e., tortilla, during controlled and ambient storage. According to Wrather et al. [25], in the field and in storage, grains of maize are the prime source of contamination by *A. flavus* infection; thus, these findings could lead to this coating being considered an alternative way to reduce *A. flavus*, together with its secondary metabolites on the grain maize (the primary source of infection by *A. flavus*) and other maize by-products such as, among others, flour, beverages, syrup, and oil (https://delmaiz.info/usos/ accessed on 15 July 2022).

Regarding the growth of the plantlet from the treated maze grains, the radicle and seedling from the maize grains coated with chitosan + nanoparticles of propolis + nanoparticles pine resin extract (T10) led to markedly lower growth in comparison with the remaining treatments. A possible explanation for this could be that the above-mentioned nanoparticles, i.e., propolis and pine resin extract, being smaller in size [26,27], were able to enter the interior of the maize grains, affecting their metabolism. On this, Lin and Xing [28] found that 2000 mg/L suspensions of Al_2_O_3_ and Zn nanoparticles considerably inhibited the root length of a variety of plant species, such as radish, rape, lettuce, cucumber, corn, and ryegrass compared to untreated ones. Other examples of this inhibitory effect on plant growth were given in the literature review performed by Khan et al. [29]. In this review, various examples of root and seedling suppression were reported when treated with different types and concentrations of metal nanoparticles, including, inter alia, tobacco, barley, cotton, cabbage, and onion.

With respect to the low levels of *A. flavus* incidence in tortillas treated with chitosan + nanoparticles of propolis + nanoparticles pine resin extract (T10), this could be associated with cellular damage to *A. flavus*. As far as we know, there are no published reports about the mode of action of nanoparticles on fungal phytopathogens; however, in observations by transmission electron microscopy shown by Sotelo Boyás et al. [30] on the plant pathogenic *Pectobacterium carotovorum* treated with nanoparticles of chitosan + thyme essential oil, there was evidence of initial cell wall damage at 6 h of treatment followed by a total cell destruction at membrane level at 48 h.

In this study, the values of weight loss and color in tortilla from coated grains were generally different from the untreated tortillas but remained within the range of acceptable quality [31].

Nanotechnology is an emerging science that has multiple applications in different fields including food microbiology [22,26,32]. In this area, the synthesis, characterization, and application of nanostructured coatings aims to preserve agricultural products of commercial interest susceptible to damage by microorganisms, such as, among others, the fungus *A. flavus*, which causes notable losses at multiple levels.

However, in addition to the benefits of NPs reported in this study, there are other questions to answer that require more evaluations, such as the adverse effects and the potential risk associated with the use of NPs. In previous studies by Hernández et al. [33], it was highlighted that nanoparticles based on plant extracts of *Byrsonima crassifolia*, α-pinene, and chitosan as source materials for the development of edible coatings, did not cause hepatoxicity, genotoxicity and cytotoxicity effects on mice at up to 2.5 mg/g concentration.

## 4. Materials and Methods

### 4.1. In Vitro Assays

#### *Aspergillus flavus* Strain

The fungus *A. flavus* was obtained from infected maize corns. Once isolated, the pathogenicity tests were performed following the morphological and molecular identification. The fungus was maintained on Czapeck-dox (Sigma-Aldrich, St. Louis, MO, USA) agar medium at 15 °C until further use.

### 4.2. Formulations and Treatment Application

#### 4.2.1. Preparation of Chitosan, Propolis, and Pine Resin Extract Solutions

The chitosan solution was prepared in accordance with the methodology of Cortes-Higareda et al. [15]. For this, medium molecular weight chitosan (Sigma-Aldrich, CAS: 9012-76-4; deacetylation degree of 75–85%) at a concentration of 1.0% was prepared by adding an equal amount (*w*/*v* 1:100) of acetic acid (Fermont Chemicals Inc. Monterrey Nuevo León, Mexico) to chitosan. This mixture was added to the total volume of distilled water and stirred overnight at ambient temperature. The solution was adjusted to pH 5.5 with 1 N NaOH solution. For propolis (Rosa Elena Dueños S.A. de C.V. Mexico city, Mexico) and pine resin extract (MS Agros. Yautepec Morelos, Mexico), 1200 µL of the liquid extract of each compound was diluted in 400 mL of 30% ethanol and 240 µL of 0.05% Tween 20 solution [20].

#### 4.2.2. Preparation of Nanoparticles of Chitosan, Propolis + Chitosan, and Pine Resin Extract + Chitosan

The chitosan nanoparticles were synthesized in accordance with the methodology implemented by Correa-Pacheco et al. [29]. Chitosan of the medium molecular weight was used at a concentration of 0.05% (*w*/*v*) and dissolved in both glacial acetic acid (0.05% *v*/*v*) and distilled water. Next, 2.5 mL of this chitosan solution was dissolved in ethanol (40 mL) using a peristaltic pump (Bio-Rad, EP-1 Econo Pump. Hercules, CA, USA) under moderate stirring. The solution obtained was placed in a rotary evaporator (Rotary Evaporator RE 300, BM 500 Water Bath, Yamato CF 300) at 40 °C and 50 rpm. The final volume of the chitosan nanoparticles was 2 mL.

For the chitosan + propolis nanoparticle formulation, the methodology employed by Correa-Pacheco et al. [31] was followed. For this, the liquid extract of propolis at 30% was dissolved in ethanol (40%) to obtain a final concentration of 0.6%. The mixture was constantly stirred for 5 min. Next, Tween 20 at a concentration of 24 µL was added, followed by a further 1 h of stirring. Finally, 2.5 mL of 0.05% chitosan was added for 10 min using a peristaltic pump. The solution was then placed in a rotary evaporator as explained above. The final volume of chitosan + propolis nanoparticles was of 2 mL. A similar methodology to that above was followed for the chitosan + pine resin extract nanoparticles, but in this case, 120 µL of 0.05% Tween 20 was added, the final volume being 2 mL.

#### 4.2.3. Formulations and Treatment Application for In Vitro Evaluations on *A. flavus*

The tested formulations were elaborated in accordance with the methodologies employed by Cortés-Higareda et al. [15] and Aparicio-García et al. [21] by changing the percentages of chitosan and the nanoparticles of chitosan, propolis, and pine resin extract (Table 1). All formulations contained 0.3% glycerol (J.T. Baker. Mexico city, Mexico). The control treatment (CT) for in vitro experiments consisted of growing the fungus only on the Czapeck-dox agar medium.

### 4.3. In Vitro Assays

#### Mycelial Growth and Spore Germination

These two variables were evaluated according to the methodology described by Cortes-Higareda et al. [15]. For this, 25 mL of each treatment was uniformly dispersed on Petri plates (6 cm in diameter) containing the Czapeck–dox culture medium. Later, 10 μL of conidia of *A. flavus* (10^5^) were placed in the center of the Petri plates and incubated at 20 °C until the control attained its maximum development. The radial mycelial growth of the fungus was measured each day in six Petri dishes of each treatment with a Truper Vernier caliper throughout 7 days of incubation. Data were evaluated as mycelial growth (cm).

For conidia germination, 10 mL of sterile water were added to four Petri dishes that belonged to each treatment. Next, conidia were scraped off the agar of each treatment. The number of spores/mL of the filtrate was adjusted to 10^5^. Of the above spore suspension, aliquots of 30 μL were placed onto six PDA disks of 20 mm diameter. Germination was terminated by adding lactophenol-safranin. One hundred observations were conducted per treatment using a Nikon ALPHAPHOT-2YS2-H optical microscope with a 40X objective. Evaluations were carried out during 0, 2, 4, 6, 8, and 10 h incubation periods. Results were expressed as a percentage of spore germination.

### 4.4. Assays on Maize Grains

#### Grain Maize Preparation, Formulations Applied, and Variables Evaluated

The maize grains were obtained from Cargill Mexico. Half of them were disinfected with 70% ethanol. Disinfected and non-disinfected grains were then punctured with a sterile needle and left to dry. Later, they were immersed (soaked) in their corresponding formulations, left to dry for 10 min, sprayed with a spore suspension of *A*. *flavus* (10^6^), and left to dry for 15 min at ambient temperature (25 °C). Next, for the variable of disease incidence (%), 10 grains per treatment were placed on Petri plates with a humid Whatman paper for 7 days at 28 °C. The applied treatments for this variable were those that caused the highest mycelial growth inhibition in the in vitro essay and were T7, T9, and T10 (Table 1 and Figure 1a). Two controls were included that consisted of wounded grains (TC1) and non-wounded grains (TC2).

For the germination and plantlet growth variables, the maize grains were dipped in the above-mentioned treatments (except for TC1) and left to dry. Twenty grains per treatment were bagged in humid polyethylene papers and kept at ambient temperature for 6 days [34]. The grains showing radicle were considered to have germinated and were measured as percentage germination (%), while the initial radicle growth of the plant was measured and expressed as plantlet growth in cm. For each of these variables, three repetitions were considered.

### 4.5. Assays on Tortilla

#### 4.5.1. Formulations Applied on Maize Grains

The selected formulations were those that, in the grain maize tests, significantly controlled *A. flavus* development, namely T9 and T10 (see Table 3), while the control treatment consisted of non-inoculated (TC2) grain maize. Before dipping in treatments (30 s), the grains were punctured with a sterile needle. They were then submerged in a spore suspension solution of 10^6^ for 15 min and left to dry for 60 min at an ambient temperature of 28 °C.

#### 4.5.2. Maize Grains Nixtamalization

Following the inoculation, the maize grains were nixtamalized (alkaline process to remove the pericarp of the grain) in accordance with the methodology of Arambula-Villa et al. [35]. The grains (1 kg) were mixed with 2% calcium hydroxide [Ca(OH)_2_] and water (2 l) (*w*/*w*) for 35 min, and cooked at 90–95 °C for 35 min. After 12 h of reposing, the grain was grounded with a mill (TORREY. Cuernavaca Morelos, México) and kneaded to a fine dough.

#### 4.5.3. Tortilla Elaboration

The dough was separated into 50 g parts and tortillas were created in a manual tortilla-maker (TORREY, México) of approximately 3 mm width and 150 mm diameter. They were cooked at 200 °C for approximately 6 min, kept in polyethylene bags, and refrigerated at 4 °C and an ambient temperature of 28 °C.

### 4.6. Evaluation of Quality Variables of Tortillas Made with Coated Maize Grains

#### 4.6.1. Humidity

For this variable, values were taken of the initial weight of six tortillas after cooking, and of the final weight after being dried in an oven at 60 °C for 40 min. The percentage of humidity was then calculated as the difference between the final and initial weight. The storage temperatures evaluated were 4 °C over 0, 3, 7,14, and 21 days and 28 °C after 0 and 3 days.

#### 4.6.2. Weight Loss

For each treatment, the weight was taken of six tortillas stored at 4 °C for 0, 3, 7, 14, and 21 days and at 28 °C for three days. Weight loss was measured as total percentage weight loss of the tortilla with respect to the initial weight.

#### 4.6.3. Rollability

For this variable, the tortilla was rolled around a 5 mm diameter wood rod, and the breakage degree was determined using the following scale: 0 = unruptured tortilla, 1–3 = partially ruptured tortilla, and 4–5 = totally ruptured tortilla [36]. This variable was evaluated 1 day after cooking and after 21 days storage. In both cases, this evaluation was carried out at 4 °C and 28 °C.

#### 4.6.4. Color

For this variable, color measurements were taken with the HunterLab (Konica Minolta. Sensing, Japan) alongside the tortilla (three measurements) at 0, 3, 7,14, and 21 days at an ambient temperature of 28 °C. The results of six tortillas were averaged and analyzed as the color difference with respect to the initial color, using the following equation:(1)$$\Delta E=\sqrt{{\left(L_{2}-L_{1}\right)}^{2}}+{\left(b_{2}-b_{1}\right)}^{2}+{\left(a_{2}-a_{1}\right)}^{2}.$$

### 4.7. Evaluation of the Presence of A. flavus Tortillas Made of Coated Maize Grains

#### 4.7.1. Disease Incidence

Samples of 20 tortillas per treatment exhibiting *A. flavus* mycelial growth were taken after 0, 3, 7, 14, and 21 storage days at 4 °C and after 0 and 3 days at 28 °C. The variable disease incidence was evaluated as a percentage of infected tortillas. To verify *A. flavus* on the samples, small portions of the symptomatic tissue were placed on Petri plates containing Czapek media and incubated for 15 days at ambient temperature. Conidia were then identified [36].

#### 4.7.2. Aflatoxin Production

To quantify aflatoxins, the methodology reported by Segura-Palacios et al. [20] was followed. Ten grams of three tortilla samples taken at 0 and 21 storage days at 4 °C was macerated and 50 mL of 70% ethanol was added and mixed in a food processor (Oster, Mexico) for 30 s at 10,000 RPM. Next, 2 mL of this solution was taken and placed in Eppendorf tubes, and then centrifuged at 12,500 RPM for 1 min. Subsequently, 100 µL of the supernatant was taken and 500 µL of developer liquid (Reveal Q+ aflatoxin kit, USA) (Lan, MI, USA) was added and mixed. Finally, 400 µL of this solution was placed in the Raptor Neogen (Lan, MI, USA) equipment for reading. The data were expressed in parts per billion (ppb) of total aflatoxins.

The selected formulations for disease incidence and aflatoxin production were: T9 and T10, control inoculated (TC1) and control non-inoculated (TC2).

### 4.8. Statistical Analysis

Treatments were arranged in a completely randomized design. Means and standard deviations were also calculated. Data were analyzed using ANOVA and where applicable, means comparison was also carried out using a Tukey test at *p* < 0.05.

## 5. Conclusions

In accordance with the planned objectives, the results demonstrated the efficacy of the formulated nanostructured coating based on a biodegradable polymer, such as chitosan incorporated with nanoparticles of both propolis and pine resin extract, in inhibiting *A. flavus* growth, both in vitro and in situ. In addition, there were no negative effects on tortilla quality; however, prior to its commercial application, additional information on the toxicity of this formulation should be generated.

## Figures and Tables

**Figure 1 molecules-27-04545-f001:**
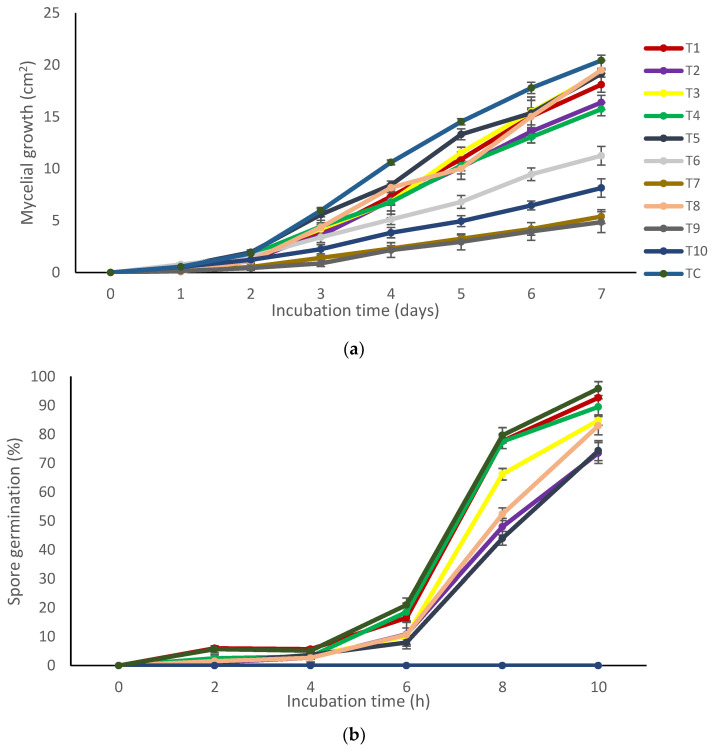
Development of *Aspergillus flavus* during a given incubation time, treated with natural-based compounds. (**a**) mycelial growth (%); (**b**) spore germination (%). Bars indicate mean standard deviations.

**Figure 2 molecules-27-04545-f002:**
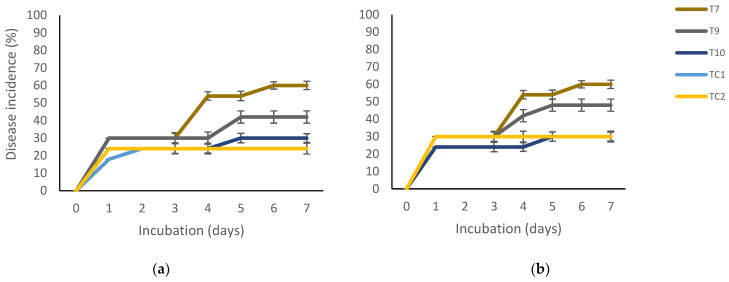
Percentage disease incidence of (**a**) disinfected and (**b**) non-disinfected maize grains, treated with based-natural coatings. T7 = chitosan + pine resin extract, T9 = chitosan + propolis + pine resin extract, T10 = chitosan + propolis nanoparticles + pine resin extract nanoparticles, TC1 = inoculated grains, TC2 = non-inoculated grains. Bars indicate mean standard deviations.

**Figure 3 molecules-27-04545-f003:**
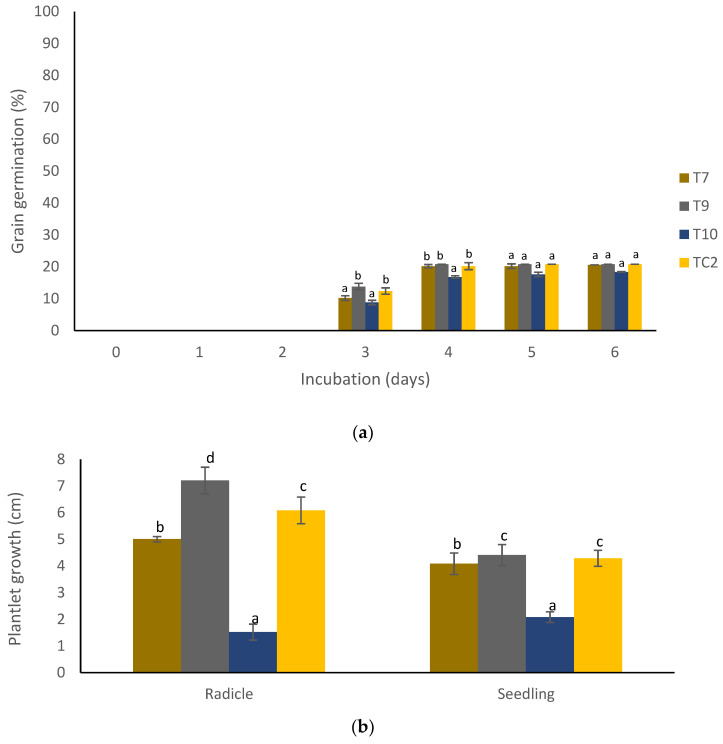
Effect of natural-based coatings on (**a**) maize grain germination and (**b**) plantlet growth. Different letters mean significant differences (*p* < 0.05; Tukey test) among treatments. Bars indicate mean standard deviations. T7 = chitosan + pine resin extract, T9 = chitosan + propolis + pine resin extract, T10 = chitosan + propolis nanoparticles + pine resin extract nanoparticles, TC2 = non-inoculated grains.

**Figure 4 molecules-27-04545-f004:**
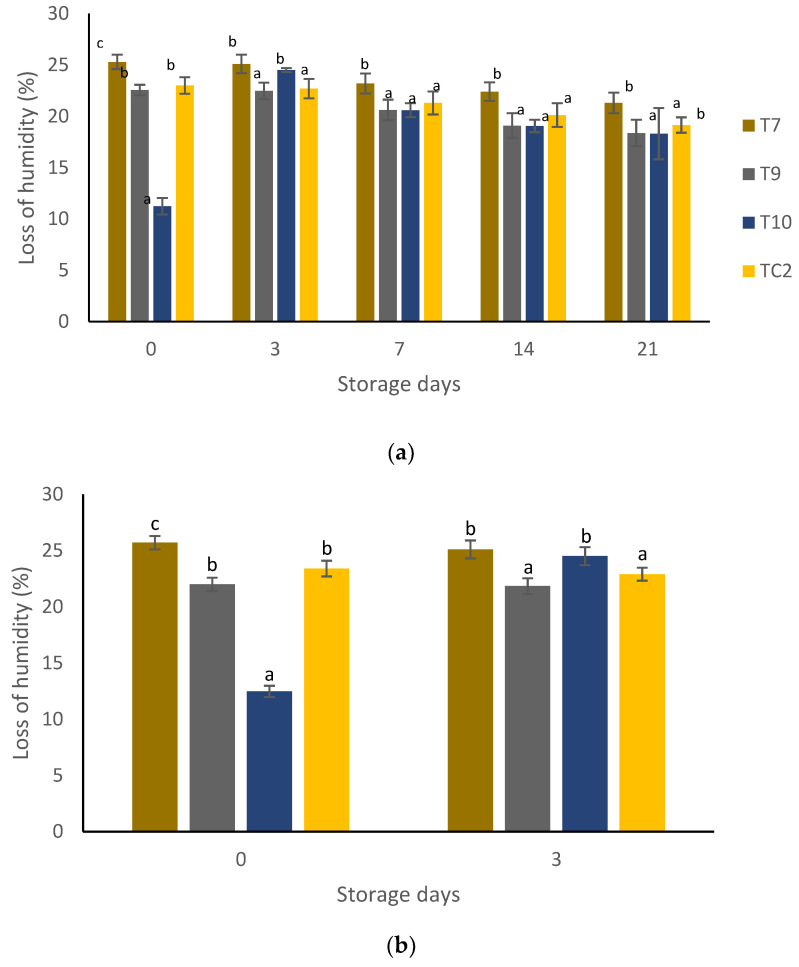
Percentage loss of humidity of tortillas elaborated with treated grains with natural-based coatings. Tortillas stored: (**a**) for 21 days at 4 °C and (**b**) three days at 28 °C. Different letters mean significant differences (*p* < 0.05; Tukey test) among treatments. Bars indicate mean standard deviations. T7 = chitosan + pine resin extract, T9 = chitosan + propolis + pine resin extract, T10 = chitosan + propolis nanoparticles + pine resin extract nanoparticles, TC2 = non-inoculated grains.

**Figure 5 molecules-27-04545-f005:**
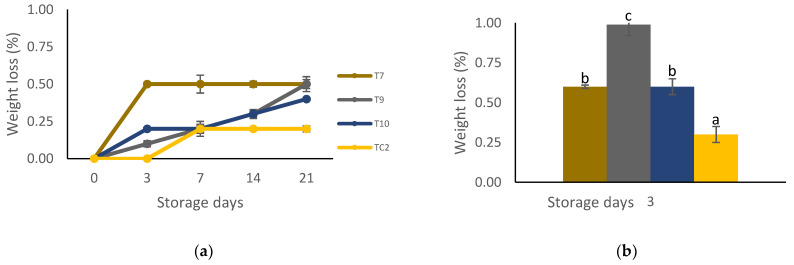
Percentage weight loss of tortillas elaborated with treated grains with natural-based coatings. Tortillas stored: (**a**) for 21 days at 4 °C and (**b**) three days at 28 °C. Different letters mean significant differences (*p* < 0.05; Tukey test) among treatments. Bars indicate mean standard deviations. T7 = chitosan + pine resin extract, T9 = chitosan + propolis + pine resin extract, T10 = chitosan + propolis nanoparticles + pine resin extract nanoparticles, TC2 = non-inoculated grains.

**Figure 6 molecules-27-04545-f006:**
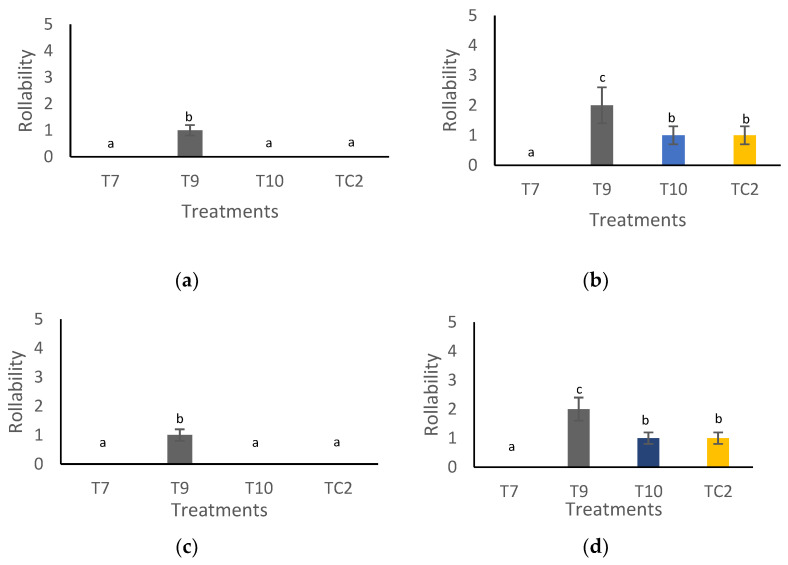
Rollability of tortillas elaborated with treated grains with natural-based coatings. Tortillas stored: (**a**) for 1 day and (**b**) 21 days at 4 °C, (**c**) for 1 day and (**d**) 21 days at 28 °C. Different letters mean significant differences (*p* < 0.05; Tukey test) among treatments. Bars indicate mean standard deviations. 0 = unruptured tortilla, 1–3 = partially ruptured tortilla, and 4–5 = totally ruptured tortilla. T7 = chitosan + pine resin extract, T9 = chitosan + propolis + pine resin extract, T10 = chitosan + propolis nanoparticles + pine resin extract nanoparticles, TC2 = non-inoculated grains.

**Figure 7 molecules-27-04545-f007:**
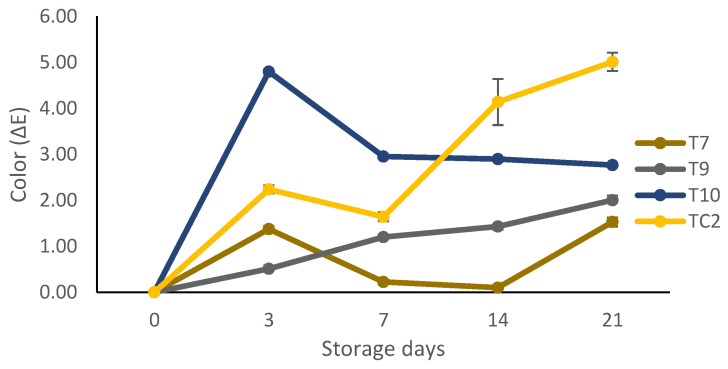
Color changes during storage of tortilla during 21 days at 28 °C, elaborated with maize grain treated with natural-based coatings. Bars indicate mean standard deviations. T7 = chitosan + pine resin extract, T9 = chitosan + propolis + pine resin extract, T10 = chitosan + propolis nanoparticles + pine resin extract nanoparticles, TC2 = non-inoculated grains.

**Table 1 molecules-27-04545-t001:** Formulations applied for in vitro studies on *Aspergillus flavus*.

Treatments	% Glycerol	% Chitosan	% Propolis	% Pine resin	% CNP	% PNP	% PRNP
T1	0.3	99.1	0.6	-	-	-	-
T2	0.3	59.1	0.6	-	-	40	-
T3	0.3	59.7	-	-	-	40	-
T4	0.3	59.1	0.6	-	40	-	-
T5	0.3	59.7	-	-	40	-	-
T6	0.3	59.7	-	-	-	-	40
T7	0.3	59.7	-	-	-	-	40.3
T8	0.3	99.7	-	-	-	-	-
T9	0.3	98.0	0.6	1.1	-	-	-
T10	0.3	59.7	-	-	-	20	20
TC	-	-	-	-	-	-	-

CNP = chitosan nanoparticles; PNP = propolis nanoparticles; PRNP = pine resin (extract) nanoparticles; CT = control treatment.

**Table 2 molecules-27-04545-t002:** Effect of inoculated grains with *Aspergillus flavus* and treated with natural-based coating on disease incidence of tortilla stored at 4 and 28 °C for a given time.

Treatments	Storage Temperature						
	Days Stored at 4 °C					Days Stored at 28 °C	
	0	3	7	14	21	0	3
	Disease Incidence (%)						
T9	0	0	0	0	0	0	100
T10	0	0	0	0	0	0	0
TC1	0	0	20	50′	50	0	100
TC2	0	0	0	15	33	0	100

T9 = chitosan + propolis + pine resin extract, T10 = chitosan + propolis nanoparticles + pine resin extract nanoparticles, TC1 = inoculated control, TC2 = non-inoculated grains.

**Table 3 molecules-27-04545-t003:** Effect of natural-based coating of inoculated grains with *Aspergillus flavus* on aflatoxin production on tortilla stored at 4 °C for 21 days.

Treatments	Storage Temperature at 4 °C (Days)	
	0	21
	Aflatoxins (ppb)	
T9	1.3 ^a^ *	1.8 ^ab^
T10	0.9 ^a^	1.0 ^a^
TC1	2.4 ^a^	3.2 ^b^
TC2	1.5 ^a^	1.7 ^ab^

* Different letters mean significant differences (*p* < 0.05; Tukey test) among treatments. T9 = chitosan + propolis + pine resin extract, T10 = chitosan + propolis nanoparticles + pine resin extract nanoparticles, TC1 = inoculated control, TC2 = non-inoculated grains.

## Data Availability

Not applicable.

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
