# Peer review of "Effect of Biodegradable Coatings on the Growth of Aspergillus flavus In Vitro, on Maize Grains, and on the Quality of Tortillas during Storage"

_molecules, 2022, doi:10.3390/molecules27144545_

Round 1

Reviewer 1 Report

A. flavus is known to cause severe crop loss and aflatoxins contamination. In this manuscript, the authors evaluated the efficacy of natural products propolis-chitosan-pine resin on the growth of A. flavus, the incidence of A. flavus on maize grains, maize tortillas, and the aflatoxins production. They found that the most effective formulation was 39.7% chitosan + 20% nanoparticles + 0.6% propolis + 1.2% pine resin extract. These results demonstrated that these coatings based on natural products, controlled A. flavus development with no adverse effects on the overall quality of the tortillas made with the coated grains. The results are technically sounded and can be used in future.

Minor concern:

The style should be improved, especally in the Results, for exmple, A. flavus should be in italic.

Author Response

Reviewer 1

  1. The style should be improved, especially in the Results, for example, A. flavus should be in italic.

Response.

The stylish of the whole text was reviewed one more time in its English edition, in some cases such as in the Introduction and Discussion sections the information was extended. All scientific names were checked, and cursives were added when necessary.

The article had already been sent for English edition

Author Response

Revisor 2

Reviewer 3 Report

In this manuscript, the anti-fungal activity of a combination of natural materials was evaluated. Authors’ work may provide the useful information on developing the environment-friendly antimicrobial methods although toxicity of formulation is a problem. Authors should re-check the whole text carefully because I found many errors in italics, capitals, spelling, and grammar.  Here are comments.

1.    In figure 2, disease incidence of TC1 is not clear. Because it is inoculated sample without treatment (positive control), about 100 % disease incidence can be expected. In figure 2 is (a) the treated sample and (b) non-treated sample? A little confusing.

2.    In figure 3, germination rate is generally very low (less than 30 %) in all conditions. Isn’t there any possibility that experimental condition for germination is not the best?

3.    The effective formula includes nanoparticles. What will be the fate of these nanoparticles in food, digested or destroyed? Authors should discuss this issue.

4.    Discussion is poorly written. Authors should improve it. Impact on taste of food, possible mechanism…

5.    There are mistakes in italics, spellings, and grammars. Please recheck the whole text.

6.    Line 297-299, method describing treatment of maize grains is not clear to me. Did authors soak the maize grains in formulation solution? Did authors dry the maize grains after immersing in formulation solutions and then inoculate those with A. flavus or inoculate A. flavus to immersed grains?

7.  Please check figure graphs. Labels of y-axis in some graphs are not properly located. It will be better to indicate scale line on x- and y-axis.

Author Response

Reviewer 3

  1. In figure 2, disease incidence of TC1 is not clear. Because it is inoculated sample without treatment (positive control), about 100 % disease incidence can be expected. In figure 2 is (a) the treated sample and (b) non-treated sample? A little confusing.

Response:

Data was checked again and is correct. When the infection is carried out artificially, it does not assure that the infection will take place in 100%. Here this is the case, and as we have experienced in other experiments with many other fungal species. Diverse defense mechanisms interfere not to let the fungi gain access to the host. In nature, the same effect happens, many intrinsic and external factors interfere to avoid the entrance of the fungi, otherwise, they would always succeed over all the horticultural products.

Letter a) corresponded to non-disinfected grains with ethanol and letter b) to disinfected ones. This has been stated in the subsection 4.4.1 of Materials and methods.  

  1. In figure 3, germination rate is generally very low (less than 30 %) in all conditions. Isn’t there any possibility that experimental condition for germination is not the best?

Response:

When studies are carried out in vitro, the methodology for seed germination followed in this present study is the most common (an additional reference was included in the section of Materials and methods to clear this). However, as mentioned above, although the low germination rate was low, this was very similar among all treatments.

The low germination was probably due that any external nutrient was given, the seed used its own resources to germinate, and as seen in the figure 3a, after 3 days of incubation, they began to deplete.

  1. The effective formula includes nanoparticles. What will be the fate of these nanoparticles in food, digested or destroyed? Authors should discuss this issue.

Response:

An extended information was included in final paragraphs of the section of Discussion.

  1. Discussion is poorly written. Authors should improve it. Impact on taste of food, possible mechanism…

Response:

Discussion was extended.

5.There are mistakes in italics, spellings, and grammars. Please recheck the whole text.

Response:

All the text was checked again. The article was sent again for English edition.

  1. Line 297-299, method describing treatment of maize grains is not clear to me. Did authors soak the maize grains in formulation solution? Did authors dry the maize grains after immersing in formulation solutions and then inoculate those with  flavusor inoculate A. flavus to immersed grains?

Response:

The methodology for the maize grains preparation has been better explained in section 4.4.1 (page 10).

  1. Please check figure graphs. Labels of y-axis in some graphs are not properly located. It will be better to indicate scale line on x- and y-axis.

Response:

All figures and graphs were checked again. The titles of the ‘X’ and ‘’Y axes were centered in Figures 2 and 3b, and Figure 5b.

Round 2

Reviewer 2 Report

I have not further comments for authors

Reviewer 3 Report

In the revised version, authors well addressed the issues that the reviewer brought up. Therefore, this manuscript is suitable for publication in present form.